# Experimental discrimination of ion stopping models near the Bragg peak in highly ionized matter

W. Cayzac[1,2,†], A. Frank[3], A. Ortner[4], V. Bagnoud[2,3], M.M. Basko[5], S. Bedacht[4], C. Bläser[4], A. Blažević[2,3], S. Busold[2,3], O. Deppert[4], J. Ding[4], M. Ehret[4], P. Fiala[4], S. Frydrych[4], D.O. Gericke[6], L. Hallo[7], J. Helfrich[4], D. Jahn[4], E. Kjartansson[4], A. Knetsch[4], D. Kraus[8,9], G. Malka[1], N.W. Neumann[4], K. Pépitone[7], D. Pepler[10], S. Sander[4], G. Schaumann[4], T. Schlegel[3], N. Schroeter[4], D. Schumacher[2], M. Seibert[4], An. Tauschwitz[11], J. Vorberger[9], F. Wagner[4], S. Weih[4], Y. Zobus[4] & M. Roth[4]

The energy deposition of ions in dense plasmas is a key process in inertial confinement fusion that determines the α-particle heating expected to trigger a burn wave in the hydrogen pellet and resulting in high thermonuclear gain. However, measurements of ion stopping in plasmas are scarce and mostly restricted to high ion velocities where theory agrees with the data. Here, we report experimental data at low projectile velocities near the Bragg peak, where the stopping force reaches its maximum. This parameter range features the largest theoretical uncertainties and conclusive data are missing until today. The precision of our measurements, combined with a reliable knowledge of the plasma parameters, allows to disprove several standard models for the stopping power for beam velocities typically encountered in inertial fusion. On the other hand, our data support theories that include a detailed treatment of strong ion-electron collisions.

[1] Université Bordeaux-CEA-CNRS, Centre Lasers Intenses et Applications, UMR 5107, 351 Cours de la Libération, 33405 Talence, France.
[2] GSI Helmholtzzentrum für Schwerionenforschung GmbH, Planckstrasse 1, 64291 Darmstadt, Germany. [3] Helmholtz-Institut Jena, Fröbelstieg 3, 07743 Jena, Germany. [4] Institut für Kernphysik, Technische Universität Darmstadt, Schlossgartenstrasse 9, 64289 Darmstadt, Germany. [5] Keldysh Institute of Applied Mathematics (KIAM), Miusskaya sq. 4, 125047 Moscow, Russia. [6] Centre for Fusion, Space and Astrophysics, Department of Physics, University of Warwick, Coventry CV4 7AL, UK. [7] CEA-Cesta, 15 Avenue des Sablières BP2, CS 60001, 33116 Le Barp, France. [8] Department of Physics, University of California, Berkeley, California 94720, USA. [9] Institute of Radiation Physics, Helmholtz-Zentrum Dresden-Rossendorf, Bautzner Landstrasse 400, 01328 Dresden, Germany. [10] STFC Rutherford Appleton Laboratory, Harwell, Oxford OX11 0QX, UK. [11] Goethe-Universität Frankfurt am Main, Max-von-Laue-Strasse 1, 60438 Frankfurt am Main, Germany. † Present address: CEA, DAM, DIF, F-91297 Arpajon, France. Correspondence and requests for materials should be addressed to W.C. (email: cayzacwitold@gmail.com).

In inertial confinement fusion (ICF), the target self-heating due to the energy loss of the 3.5 MeV α-particles born from deuterium–tritium fusion reactions must dominate all loss processes. Precise understanding of this process is thus required for modelling the diverse plasmas within the dynamic environments of a burning fusion pellet, especially for ignition and the launch of a thermonuclear burn wave leading, ultimately, to an energy gain[1]. Ion stopping in a plasma is also crucial for target heating schemes using ion beams as main drivers, like heavy-ion fusion[2] or ion-driven fast ignition[3]. The relevant physical quantity is the stopping power of the projectile ions due to Coulomb interactions with the plasma particles, that is, $dE/dx$. The stopping power peaks when the projectile velocity $v_p$ reaches the thermal velocity of plasma electrons and ions, respectively, that is for

$$v_p \approx v_{th}^{e,i} = \sqrt{\frac{3k_B T_{e,i}}{m_{e,i}}}. \tag{1}$$

Here, $T_{e,i}$ and $m_{e,i}$ are the temperature and the mass of the plasma electrons and ions respectively. The contributions from plasma ions are negligible except if $v_p \approx v_{th}^i$, that is, for very low beam velocities or very hot, already burning ICF plasmas. The electronic stopping power reaches a maximum in the velocity range $v_p \approx v_{th}^e$, causing the Bragg peak where a considerable part of the beam energy is deposited in a small volume.

For fast projectiles $\left(v_p \gg v_{th}^e\right)$, the stopping power is dominated by weak, long-range interactions with the plasma electrons. It can then be described by perturbative theories[4–7], which agree well with data in this regime[8–12]. In contrast, there is a sparse database for beam velocities $v_p \approx v_{th}^e$ and the few experiments performed[13–15] were not conclusive mainly due to uncertainties in the plasma temperature. Around the Bragg peak, stopping theories that include a detailed treatment of close collisions between the projectiles and the plasma electrons[16–18] predict 30–50% smaller energy loss than the standard perturbative models[16,17,19]. These theoretical uncertainties are critical for the prediction of α-particle heating in ICF[20], where $v_p \leq v_{th}^e$ holds for most of the range. Indeed, a reduced α-particle stopping leads to a range enhancement by almost the same factor. This can affect the ignition threshold in the dense main fuel as well as the energy deposition inside the hot spot of a burning deuterium–tritium plasma that initiates the thermonuclear burn wave.

We test the validity of stopping-power models in a laser-generated plasma, where the relevant target parameters of electron coupling and degeneracy are similar to the ones in an ICF plasma. The discrepancies between the predictions of different stopping models reach up to 30% in such a plasma for ions probing the maximum of the stopping power and, thus, can be tested experimentally[19]. We present measurements for nitrogen ions probing a laser-induced carbon plasma at beam velocities with $v_p \approx v_{th}^e$. These data provide a discriminating test of stopping-power theories in a parameter range relevant to fusion plasmas.

## Results
**Experimental set-up**. The experiment was carried out at the GSI Helmholtzzentrum für Schwerionenforschung GmbH, Darmstadt, Germany. Here, two high-energy lasers, PHELIX and nhelix[21], were used to create the plasma that was later probed with a pulsed ion beam from the UNILAC accelerator (Fig. 1). A 100 μg cm$^{-2}$ carbon foil is heated from both sides by two laser beams, leading to full target ionization after 6–7 ns with free electron densities of $n_e \approx 5 \times 10^{20}$ cm$^{-3}$ and electron temperatures of $T_e \approx 150$ eV (refs 11,19). These plasma

parameters correspond to nearly ideal and nondegenerate target conditions, as indicated by the respective dimensionless parameters for electron coupling $\Gamma$,

$$\Gamma = \frac{e^2}{a_e k_B T_e} \approx 0.01, \tag{2}$$

and degeneracy $\Theta$,

$$\Theta = \frac{k_B T_e}{E_F} \approx 550, \tag{3}$$

with $a_e = (4\pi n_e/3)^{-\frac{1}{3}}$ being the average distance between the electrons and $E_F$ the Fermi energy of the free electron gas. These conditions are similar to burning ICF plasmas ($\Gamma \approx 0.01$ and $\Theta \approx 30$ at $n_e \approx 10^{25}$ cm$^{-3}$ and $T_e \approx 5$ keV).

For probing the plasma at the Bragg peak, the ion bunches of originally $E_p = 3.6$ MeV per nucleon energy are degraded through a 41 μm thick carbon foil, resulting in a mean ion energy in the range $E_p = 0.586 \pm 0.016$ MeV per nucleon. This beam energy corresponds to $v_p/v_{th}^e \approx 1.2$ and to $v_p/v_{th}^i \approx 175$, hence allowing to probe the electronic stopping at its maximum while the contribution of plasma ions remains negligible. Moreover, the Coulomb parameter $\eta = Z_b e^2/(\hbar v_r)$, $Z_b$ being the beam charge and $v_r$ the relative velocity between the projectile ions and the plasma electrons, is roughly unity. This indicates a significant beam–plasma coupling and, thus, the importance of strong non-perturbative collisions, while quantum diffraction effects still play a role. The energy loss of the beam ions is measured using the time-of-flight (TOF) method, with a semiconductor detector based on chemical vapour deposition (CVD) diamond[22] (Fig. 1c). Our detector permits an energy resolution of $\Delta E_p \approx 70$ keV ($\Delta E_p/E_p \approx 1\%$), which is much smaller than the differences between the predictions of several stopping-power models[19].

**Simulations of the target and beam properties**. The plasma conditions were simulated with the two-dimensional (2D) hydrodynamic code RALEF2D (ref. 23) for the times $t = 0$–15 ns after the beginning of the target heating. In this range, the temperature remains sufficiently high to guarantee $v_p \approx v_{th}^e$. The plasma ionization is deduced by post-processing the density and temperature profiles with the FLYCHK code[24]. This determines the free electron density, whose distribution is illustrated in Fig. 2c for $t = 7$ ns. The density profiles were benchmarked against time-resolved laser interferometry measurements[25]. An example of raw data is shown in Fig. 2a for $t = 11$ ns, and examples of measured profiles are presented in Fig. 2b for $t = 7$ and 11 ns. Comparing the data with the simulations for the times of the measurements, that is $t = 5$–15 ns, reveals maximum discrepancies by a factor of two. However, the predictions are in much better agreement for the majority of times. Complementary simulations indicate a corresponding maximum uncertainty in the plasma temperature of $\pm 40$ eV.

The simulated plasma areal density along the ion trajectory $\rho R = \int \rho(x)dx$, with $\rho$ being the mass density, is benchmarked against previous energy-loss measurements for fast ions with known stopping power[11] (Fig. 2d). The areal density remains constant for 0–7 ns due to one-dimensional (1D) plasma expansion and later decreases with the three-dimensional (3D) expansion, reducing the energy loss. The good overall agreement of the experimental data with all stopping predictions validates the simulated time evolution of the important areal density.

The effective beam charge state in the plasma, that is a crucial parameter for the stopping power, was calculated using a Monte-Carlo code based on projectile electron loss and capture rates[11,19] as well as the models by Gus'kov et al.[26] and Kreussler et al.[27]. These models all agree with the energy-loss measurements of

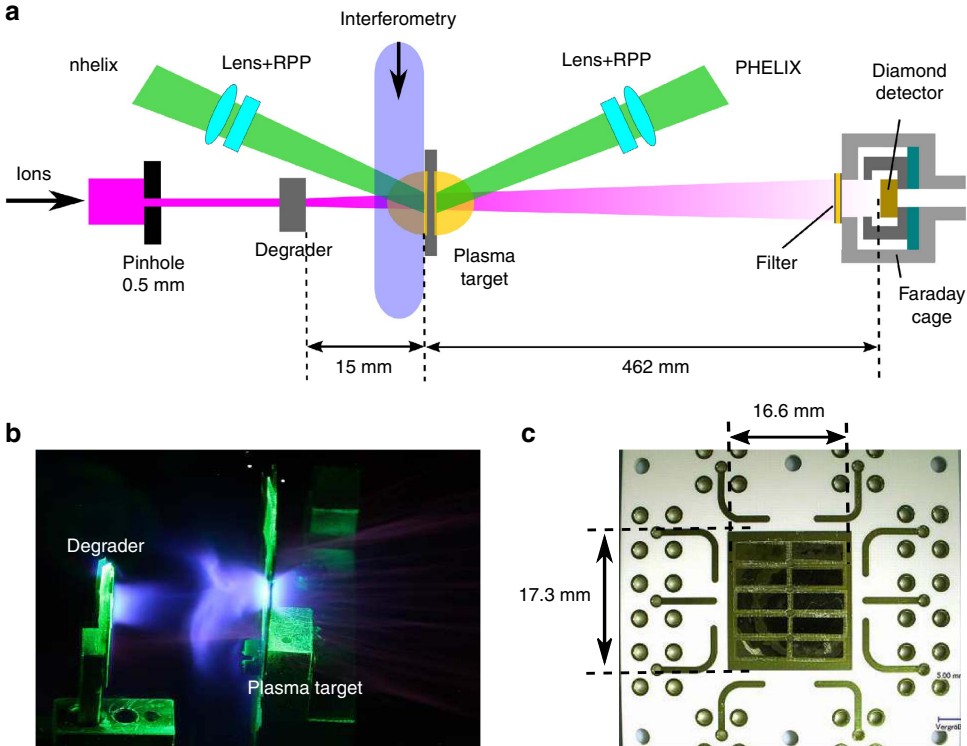

**Figure 1 | Experimental set-up. (a)** Schematics of the experimental set-up. Two high-energy laser beams are focused on a 100 µg cm$^{-2}$ thick carbon foil with a 1 mm diameter obtained by beam smoothing using RPP. The plasma electron density is measured with a laser interferometry diagnostic. The ion beam is collimated through a 0.5 mm diameter pinhole and degraded through a carbon foil before interacting with the plasma, and it is detected after a 462 mm TOF distance. **(b)** Time-integrated picture of one plasma shot registered with a digital camera. **(c)** Picture of the TOF detector displaying the ten diamond samples mounted on their printed circuit board.

Fig. 2d in the high-velocity region. The time evolution of the effective charge in the plasma applying these models is shown in Fig. 3a and compared with the mean charge state in the solid target ($Z_{sol} = 4.88$ according to Ziegler et al.[28]). The effective charge is enhanced in the plasma compared to the solid target, mainly due to a smaller electron capture cross section of the projectiles. The increase reaches from half to a full charge state depending on the model, which contributes by up to 19 or 45%, respectively, to the stopping-power enhancement.

The simulated energy loss $\Delta E_{sim}$ is calculated as the integral of the stopping power along the ion trajectory

$$\Delta E_{sim} = -\int \frac{\partial E}{[\rho(x)\partial x]}[\rho(x)\mathrm{d}x], \qquad (4)$$

where the stopping power is expressed as an energy loss per unit of areal density. Each energy-loss value is averaged over the plasma parameters in a 5.5 ns range corresponding to the experimental bunch duration.

**Experimental results**. The experimental energy loss $\Delta E_{exp}$ is determined from the time shifts in the detector signals due to ions penetrating the plasma compared to undisturbed ions, for different probing times within the relevant interval $t = 0$–15 ns. The data for the experimental energy loss, compiled from eight different shots, are presented in Fig. 3b–d. Each value is normalized to the energy loss in the corresponding solid target (measured to be in the range 0.83 ± 0.03 MeV) for smoothing out the few per cent shot-to-shot differences in the target thickness. The error bars correspond to one s.d. (1$\sigma$) of the uncertainty in the time shifts. $\Delta E_{exp}$ is enhanced by up to 50% compared to the solid state, both due to a more efficient momentum transfer of the

projectile ions to the plasma free electrons and due to the increase in the beam charge state.

The data are compared with the predictions of the Li–Petrasso (LP) stopping model[7] which here stands for the standard stopping approaches and gives similar results[19] to the standard stopping model by Deutsch[4] or dielectric approach[5,6]. Furthermore, we compare to the model by Brown–Preston–Singleton (BPS)[18] and the T-matrix formulation employing a velocity-dependent screening length (TM)[17], which both include a detailed treatment of close binary collisions as well as quantum diffraction effects. In Fig. 3b, the data are compared with the LP and TM predictions applying the three considered models for the effective charge. Only the TM stopping model combined with either the Gus'kov or Kreussler charge model agrees with the measurements, while other simulations overestimate the energy loss by at least 20%. For simplicity, only the Gus'kov model is used in the following. In Fig. 3c, the data are additionally compared with the BPS model, which predicts values very similar to the TM model. Moreover, the effect of the transversal beam as well as plasma profiles is illustrated. For each model, the shaded area indicates the energy-loss reduction (up to 10%) when considering the 2D plasma profile with a transversal decrease in areal density (bottom line) instead of the 1D profile along the ion axis (upper line).

In Fig. 3d, we finally display the global error on $\Delta E_{sim}$ due to uncertainties in the electron temperature and in the free electron density, using the LP and the TM stopping-power models. Taking as respective uncertainties the maximum differences resulting from the simulation benchmarking against the density measurements, we performed complementary energy-loss calculations, first for temperature profiles $T'_e = T_e \pm 40$ eV, and second for free electron density profiles

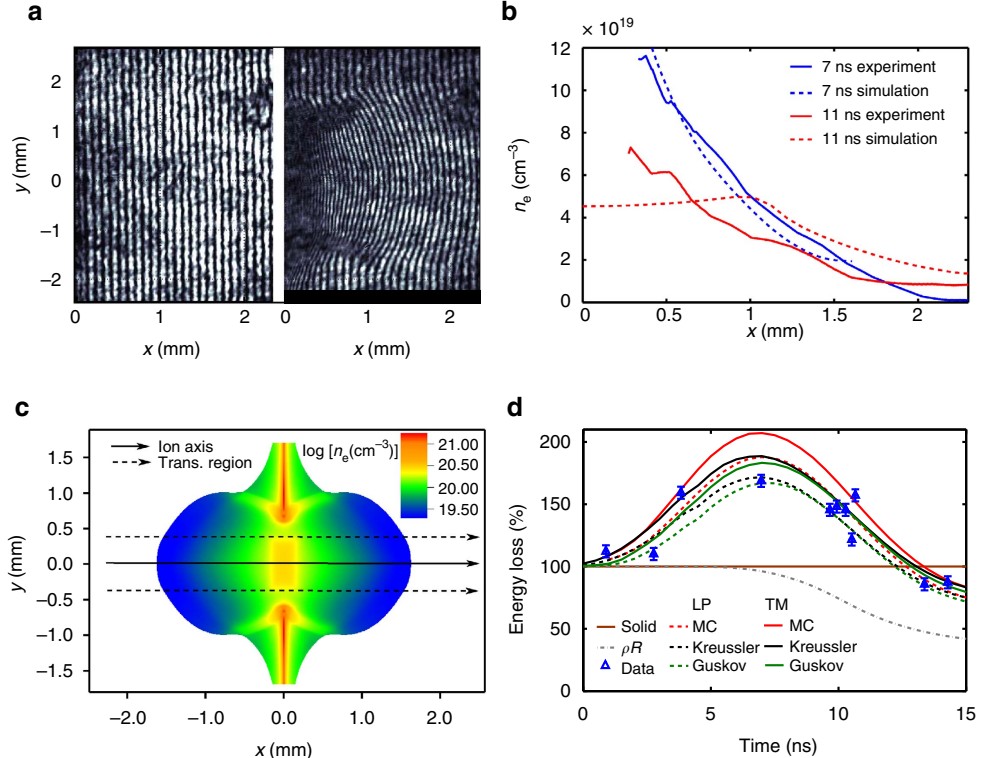

**Figure 2 | Plasma characterization. (a)** Raw interferometry data for the plasma for $t = 11$ ns (right), compared with the reference measurement (left). The target surface is located on the left side of each picture. **(b)** Comparison between measured and simulated free electron density ($n_e$) profiles along the ion axis ($x$) for $t = 7$ and 11 ns respectively. Experimental error bars, of approximatively 20%, are not represented. **(c)** 2D map of the simulations of the free electron density ($n_e$) for $t = 7$ ns, in units of cm$^{-3}$ and represented in logarithmic scale. The solid arrow stands for the ion axis (1D plasma profile) used in the energy-loss calculations throughout the paper. The dotted arrows delimit the transversal region considered for the energy-loss calculation using a 2D plasma profile in Fig. 3c. **(d)** Energy-loss measurements for the same plasma versus the probing time, for argon projectile ions at a velocity ratio $v_p/v_{th}^e \approx 3.1$[11]. The data are compared with the predictions of the LP and the TM stopping-power models applying a Monte-Carlo description as well as the Gus'kov and the Kreussler models for the effective projectile charge state. The energy loss is normalized to its value in the solid target (100%), as well as the plasma areal density ($\rho R$). The error bars correspond to one s.d. ($1\sigma$) of the uncertainty in the time shifts of the signals obtained from the TOF measurements.

$n_e' = n_e/2$ and $n_e' = 2 n_e$, respectively. However, the areal density was kept as measured here. As the stopping power decreases with temperature and with density in the considered parameter range, the upper boundaries of the energy-loss calculations correspond to the lower temperature and density, while the lower energy-loss boundaries are obtained for the highest allowed temperature and density. The uncertainty in temperature leads to a maximum error of $\pm 15\%$, and the uncertainty in density, a maximum error of $\pm 10\%$ on $\Delta E_{sim}$. Assuming these contributions as independent, a maximum global error on $\Delta E_{sim}$ of around $\pm 15\%$ is estimated. This error remains smaller than the discrepancy between the predictions of the LP model and the experimental data, while the TM predictions are within the $1\sigma$ experimental error bars. This demonstrates that the plasma parameters are known with a sufficient precision to discriminate between these stopping-power models with our measurements. We are able to distinguish between the models because the considered ion beam and plasma parameters lead to large stopping powers and also strong beam–plasma coupling, which imply important differences between the predictions. The LP model and, thus, also other perturbative models, systematically overestimate the energy loss by 20–25%, that is, outside the $1\sigma$ error bars. In contrast, the TM and the BPS predictions prove to be good fits to the experimental data. Hence, in the studied parameter range, our results disprove the standard perturbative models, while they support the TM and the BPS models. In addition, the results also support the beam charge-state formalisms by Gus'kov and Kreussler.

Consequently, our data provide a conclusive test of stopping-power predictions for the velocity range around the Bragg peak by discriminating between different classes of theories in highly ionized plasmas to better than $1\sigma$. Our results show that even for nearly ideal and nondegenerate plasma conditions, close collisions between the projectiles and the plasma electrons modify the stopping power for slow- to medium-velocity ions significantly. Therefore, widely applied models for the stopping power that are essentially based on perturbation theory, fail to reproduce our experimental data and are ruled out by our measurements. Instead, approaches that include a full description of binary collisions agree with the data. This means that the dominant Coulomb collisions need to be more accurately modelled in the important velocity range around the Bragg peak. This finding has also strong implications for other transport and relaxation properties like temperature equilibration[29] or thermal and electrical conductivity[30] where close collisions play a similarly important role.

Future developments of these experiments include energy-loss measurements with protons and α-particles. These projectiles, directly relevant for ICF, are essentially fully stripped in a plasma, which removes the need to model the beam charge state. Thus, theories can then be compared even more directly with measured data and more precise insights into the physics of collisional processes can be gained. Other techniques are required to obtain data for even slower particles at the end of the range. Furthermore, modelling the complex physics of self-heating and

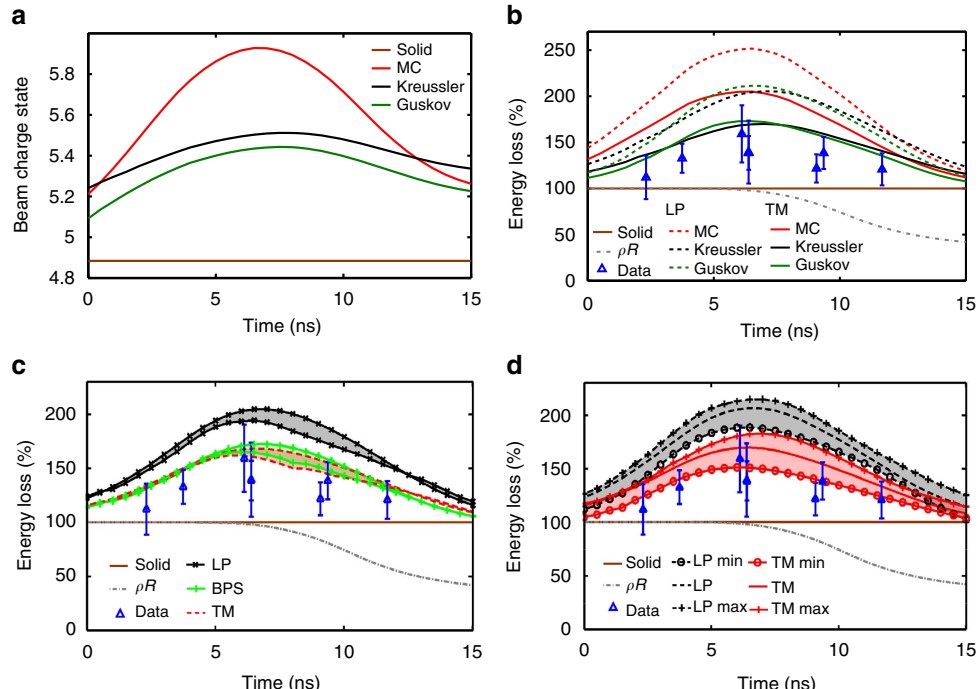

**Figure 3 | Energy-loss results.** (**a**) Effective charge state of a nitrogen bunch in the plasma versus time according to a Monte-Carlo description as well as the Kreussler and the Gus'kov models, compared with the mean charge state in the solid target $Z_{sol} = 4.88$. All values are averaged over the ion trajectory through the target. (**b**) Measured energy loss as a function of the bunch probing time and normalized to its value in the solid target (taken to be 100%) compared with the predictions of the LP and the TM stopping-power models applying the Monte-Carlo, Kreussler and Gus'kov projectile charge models respectively. The simulated target areal density ($\rho R$), also normalized to its value in the solid target, represents the 3D plasma expansion dynamics. (**c**) Measured energy loss compared with the predictions of the LP, TM as well as BPS stopping-power models using the Gus'kov projectile charge model. The shaded areas show the differences between calculations considering the 1D (upper lines) or 2D plasma profile (bottom lines) respectively (cf. Fig. 2c). (**d**) Measured energy loss compared with calculations for the LP and TM stopping-power models using the Gus'kov projectile charge model, corresponding to the originally simulated density and temperature profiles (LP; TM), densities $n'_e = n_e/2$ and temperatures $T'_e = T_e - 40$ eV (LP max; TM max) as well as densities $n'_e = 2\,n_e$ and temperatures $T'_e = T_e + 40$ eV (LP min; TM min), respectively. The shaded areas thus illustrate the maximum error in the energy-loss calculation due to uncertainties in the plasma parameters. Due to time averaging over the 5.5 ns bunch, the beam charge state in **a** as well as the energy loss in (**b**–**d**) for $t = 0$ ns, are already larger than their respective values in the solid target. The error bars on the energy loss correspond to one s.d. ($1\sigma$) of the uncertainty in the time shifts in the detector signals.

burn waves in ICF fuels requires additional experimental data at the Bragg peak for degenerate and moderately coupled plasmas.

## Methods

**Set-up.** The nitrogen beam had an original ion energy of 3.6 MeV per nucleon. The bunches had a frequency of 36 MHz, featured Gaussian temporal profiles with a duration of 5.5 ns at full width at half maximum (FWHM) and contained about one thousand ions each. Nitrogen was chosen as projectile as the lightest ion species available at the time of the experiment, in order to simplify the beam charge-state distribution in the plasma insofar as possible.

The PHELIX and nhelix beams are generated with wavelengths of 1,053 and 1,064 nm respectively, amplified in similar Nd:glass chains and frequency-doubled using, respectively, potassium dihydrogen phosphate and deuterated potassium dihydrogen phosphate crystals before focusing on the target. The beams have a pulse duration of 7 ns (FWHM) and their energy was measured to be 28 ± 4 J. They are spatially smoothed using random phase plates (RPP)[31], manufactured by Scitech Precision Ltd., a spin-off from Rutherford Appleton Laboratory, Didcot, UK. The RPP create a top-hat focus profile with a 1 mm diameter on the target, resulting in a laser intensity on target $I \approx 5 \times 10^{11}$ W cm$^{-2}$ and in transversally uniform plasma parameters within this area. Moreover, 3D plasma expansion effects on the energy-loss measurements are minimized by probing the target only in its central region through the use of a 0.5 mm diameter pinhole located a few centimetres in front of the degrader. The two laser beams are synchronized with each other and with the ion beam with a precision better than 1 ns, which corresponds to the maximum jitter of the system.

The interferometry beam is generated by an oscillator synchronized with the nhelix oscillator. A detailed description of the interferometer can be found elsewhere[25].

All the used foils (targets, degraders and detector filters) were produced at GSI Target Laboratory and were calibrated, along with the detector, in several preliminary experimental campaigns.

**Targets.** The targets for plasma generation are self-supporting carbon foils obtained by resistance evaporation under high vacuum[32]. They had a mass density of $\rho \approx 1.3$ g cm$^{-3}$ and an initial areal density in the range 96 ± 5 µg cm$^{-2}$, the areal density of each foil being known with a precision of ± 1 µg cm$^{-2}$.

The degraders had a density $\rho = 1.84$ g cm$^{-3}$ and were produced by rolling carbon foils down to a thickness approaching the required value of 41 µm predicted by simulations using the SRIM/TRIM code[28]. These simulations predict a straggling of the beam energies of 5% at $1\sigma$ (± 30 keV per nucleon), corresponding to a straggling of the beam velocities smaller than 3% and an angular straggling of ± 3°. The degrader is positioned 15 mm in front of the target, which both ensures a free path for the heating lasers and limits the transverse broadening of the ion beam when probing the plasma. Monte-Carlo simulations show that the uncertainty on the TOF measurement induced by the energy straggling of the beam is smaller than 1%. The degrader was systematically destroyed by the plasma emission and expansion from the target and had to be changed after each shot. The mean ion energy was determined from the TOF measurement with a precision better than ± 0.010 MeV per nucleon and it remained in the range 0.586 ± 0.016 MeV per nucleon over the various shots. The stopping-power variations due to the beam energy straggling and due to the beam energy variation from shot-to-shot remain limited to 1%, which can be neglected compared to the uncertainties in the plasma parameters. The ablation of the degrader surface, caused by the plasma X-ray emission and subsequent expansion, was monitored with an optical streak camera and proved to only be significant later than 25 ns after the beginning of the laser heating of the target, which does not affect the energy-loss measurements, performed in the first 15 ns.

**TOF diagnostic.** The TOF detector is based on ten identical polycrystalline CVD diamond samples, manufactured by Diamond Materials GmbH, a Spin-Off from Fraunhofer Institute IAF in Freiburg, Germany. They were metallized at GSI Target Laboratory with the help of a magnetron-sputtering device, with layers of, from inwards to outwards, 100 nm titanium, 30 nm platinum and 20 nm gold. Each

diamond has a thickness of 13 μm and an area of $8.2 \times 3.3 \, mm^2$, including a $7.2 \times 2.3 \, mm^2$ metallized zone. These dimensions result in a detector time constant of 2.8 ns when amplifying the signals with 50 Ω-matched broadband amplifiers, which was measured from single-particle signals obtained from an α-particle source. The total detection area of 166 $mm^2$ is large enough to collect about 20% of the ions at the beam focus position, which guarantees sufficient signal amplitudes for a quantitative energy-loss analysis. The detector is located 462 mm behind the target, which is sufficiently short to avoid bunches to overlap due to the distribution of energies in the beam. The time resolution of the detector of 0.25 ns, that is, the maximum precision on the determination of the centre of mass of the ion bunch signals, combined with a 462 mm TOF distance, implies an energy resolution of $\Delta E_p \approx 70$ keV.

The detector is shielded in two ways, against plasma-emitted X-rays and electromagnetic pulses, respectively. First, an X-ray filter is positioned in the ion path over the detector aperture at a 20 mm distance from the diamond surface. The filter is a self-supporting foil made of a 1,500 μg cm$^{-2}$ thick gold layer and an additional 400 μg cm$^{-2}$ thick carbon layer, produced by resistance evaporation under high vacuum. The areal density of each foil is known with a precision better than 5%. Through the filter, most X-rays are absorbed while the ions are transmitted, although with a significantly reduced energy. The resulting TOF variation is taken into account in the data analysis. The filter was systematically damaged and had to be changed after each shot. Second, the detector and the signal transmission line are enclosed in a Faraday cage for disconnecting their electrical ground from the target chamber. Applying this shielding enabled to drop the detector saturation time down to ∼20 ns, which is shorter than the ion TOF of nearly 50 ns from the plasma to the detector. Therefore it guarantees that the signals of the bunches that passed the plasma are not significantly affected by electromagnetic radiation.

**Data analysis.** For each laser shot, the ion bunch signals are registered during a time interval of at least 40 μs that is selected by using a chopper magnet. This time range starts with signals of ions that penetrated the initial solid target, goes through the complete plasma evolution with the destruction of the target and of the degrader, and ends with undisturbed bunch signals after the plasma has completely expanded. Note that the remaining filter has a negligible influence on the 3.6 MeV per nucleon energy ions. The undisturbed signals correspond to a zero energy loss and serve as a reference measurement. The comparison of the centres of mass of the signals with those of the reference signals gives the total mean time shift $\Delta t$ of the bunch ions (through the degrader, the target and the filter). The total mean energy loss $\Delta E$ experienced by ions in each bunch is then determined using

$$\Delta t = L \left( \frac{1}{v_p(\Delta E = 0)} - \frac{1}{v_p(\Delta E)} \right), \quad (5)$$

where $L$ is the TOF distance, $v_p(\Delta E = 0)$ is the reference ion velocity and $v_p(\Delta E)$ is the final ion velocity (after the filter). Having determined the final ion energy from the total bunch shifts, and knowing the thickness of the solid target and of the filter foil (with a precision of 1 and 5%, respectively), the ion energy after the degrader as well as the energy loss in the solid target are deduced using simulations with the SRIM code. The energy loss in the plasma is obtained from the time shift of the first bunch that passed through the plasma, knowing the corresponding shift in the solid target. The data analysis method is illustrated in Fig. 4 on raw detector signals from one experimental shot.

As the relevant plasma probing time is limited to the first 15 ns and the bunch period is 27.7 ns, only the first bunch that passed the plasma, purposely timed in relation to the lasers within the 15 ns interval of interest, is used for the energy-loss analysis. For each shot, the mean bunch probing time through the plasma, that is, the abscissa of the corresponding data point in Fig. 3b–d, is obtained by deconvoluting the response function of the detector, approximated as a decaying exponential function, from the Gaussian temporal ion bunch signals.

The error bars displayed on the energy-loss data correspond to one s.d. of the uncertainty in the value of the time shift of the bunch of interest. For clarity purposes, horizontal error bars of ±1 ns corresponding to the maximum jitter between the lasers and the ion beam are not represented as they do not modify the data interpretation. The uncertainty in the time shift of the bunches penetrating the solid target is obtained by a Gaussian error propagation including the s.d. of the centres of mass of the undisturbed signals (≈0.3 ns) as well as of the signals penetrating the solid targets (≈0.2–0.3 ns), the uncertainty in the measured TOF distance (2 mm, which corresponds to 0.2 ns in terms of time shift), the uncertainty in the initial areal density of the target (1 μg cm$^{-2}$, which corresponds to 0.03 ns in terms of time shift) and the uncertainty in the filter thickness (5%, leading to an uncertainty of 0.04 ns in the time shift). These various contributions result in a global error on the time shift in the solid target in the range 0.41–0.75 ns, depending on the shot. Hence, the mean energy loss in the solid target, determined to be in the range 0.83 ± 0.03 MeV, has a mean error in the range 0.13–0.27 MeV depending on the shot.

For the bunch relevant for the energy-loss analysis, we add the error contribution due to noise originating from electromagnetic perturbations on the detector, determined from the error in fitting the signal (0.1–0.5 ns). The total mean uncertainty in $\Delta t$ is in the range 0.42–0.78 ns, which corresponds to energy-loss error bars of 0.13–0.28 MeV, or 15.3–34.1% of the energy loss in the

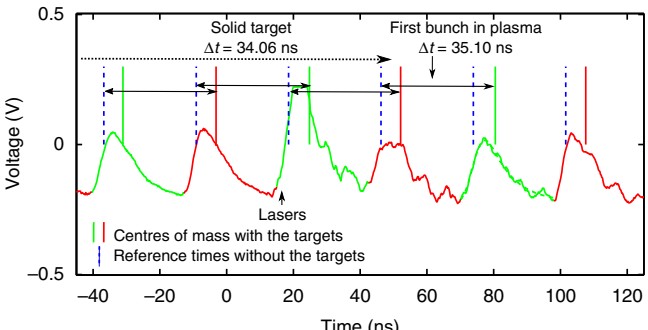

**Figure 4 | Raw data from the TOF detector.** Sample of the detector signals through the solid target and through the plasma, for the shot corresponding to the data point at $t = 6.4$ ns in Fig. 3b–d. The time shift ($\Delta t$) of each bunch is determined between the centre of mass of this bunch (plain vertical bar) and the associated reference time (dotted vertical bar). The shown reference times are extrapolated from the period of the undisturbed ion bunch signals few tens of μs after the plasma expansion. The average measured time shift in the solid foils (degrader, target and filter) is $\Delta t = 34.06 \pm 0.45$ ns. Knowing the initial areal density of the target of $95 \pm 1$ μg cm$^{-2}$ as well as of the filter, the projectile energy after the degrader is deduced to be $E_p = 0.576 \pm 0.005$ MeV per nucleon, the time shift through the solid target $\Delta t = 2.39 \pm 0.45$ ns and the energy loss in the solid target $\Delta E = 0.82 \pm 0.14$ MeV. Consequently, the first ion bunch that passed through the plasma features a time shift in the plasma target of $\Delta t = 3.43 \pm 0.47$ ns (the global shift being $\Delta t = 35.10$ ns). This corresponds to an energy loss in the plasma of $\Delta E = 1.14 \pm 0.15$ MeV, or $139.5 \pm 18.3\%$ normalized to the value in the solid target.

solid target. Hence, the experimental error on the energy loss is significantly larger than the detector's energy resolution of 0.07 MeV and does not allow a fine benchmarking of theories. However, it still enables to clearly discriminate two classes of stopping-power theories and, in particular, to disprove the perturbative models.

**Simulations.** The energy-loss simulations are described in detail elsewhere[19]. Here, however, the energy loss is exclusively calculated assuming that the plasma is not in local thermodynamical equilibrium. Consequently, we use the FLYCHK code with its collisional-radiative model to describe the electron exchange dynamics in the plasma. For the calculation of the projectile charge state, in addition to the previously used Monte-Carlo code to predict the mean beam charge[11,19], we use two alternative descriptions that appear to be more accurate for low projectile velocities. The model by Gus'kov et al.[26] calculates the effective beam charge state in plasma employing a formalism similar to the semi-empirical formula by Ziegler et al.[28] for the mean charge state in a solid target, with inclusion of the influence of the target electron motion. This formulation is then extended to plasmas as a function of the plasma density and temperature as well as of the beam velocity, with averaging of the thermal electron motion over the Fermi-Dirac plasma electron distribution. The model by Kreussler et al.[27] uses a similar method to calculate the mean beam charge state. For relatively light ions as nitrogen, the mean and the effective beam charge state are very similar, therefore we speak only of effective charge throughout the paper, which is the quantity directly related to the stopping power.

The energy-loss calculation with the 2D plasma profile is performed by considering the contributions of all ions within the detector solid angle, that is, those penetrating the plasma within a radius of 400 μm from its central axis. Over this transversal profile, the plasma areal density decreases by 20–30%. This leads to a correspondingly smaller energy loss for ions passing near the plasma edge, that is estimated by averaging the Gaussian transversal ion beam profile obtained from the SRIM simulations over the transversal areal density profile of the plasma.

As a general remark to the simulations, note that due to the relatively low number of ions per bunch (≈1,000), each bunch deposits ∼1 GeV energy inside the plasma, which is negligible compared to the global thermal plasma energy (≈$10^{14}$ GeV cm$^{-3}$ for a density of $n_e \approx 5 \times 10^{20}$ cm$^{-3}$ and a temperature of $T_e \approx 150$ eV). Therefore, the energy loss of the projectile ions in the plasma does not influence the plasma evolution.

**Data availability.** The data that support the findings of this study are available from the corresponding author upon reasonable request.

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

## Acknowledgements

We gratefully thank the teams of the UNILAC accelerator and of the PHELIX laser at the GSI Helmholtzzentrum für Schwerionenforschung for their expert support in performing the experiment, as well as of the Target Laboratory and of the Detector Laboratory of GSI in preparing it. We thank V. Tikhonchuk for his support in the experiment planning. This work was supported by Fonds de Coopération Interrégionale between the Région Aquitaine and the Land Hessen, Bundesministerium für Bildung und Forschung (BMBF), the Helmholtz International Center for FAIR (HIC4FAIR) as well as EUROfusion Consortium (toIFE programme, Grant agreement No 633053). The views and opinions expressed therein do not necessarily reflect those of the European Commission.

## Author contributions

W.C., A.F., A.O. and A.B. designed the experiments. W.C. and A.K. developed the CVD diamond detector. V.B., J.H., D.S. and A.O. were lead experimenters for the PHELIX and nhelix laser facilities as well as for the interferometry diagnostic. W.C. Analysed the data. M.M.B. and A.T. performed the RALEF2D simulations. W.C., J.V. and D.O.G. carried out the ion-stopping calculations. K.P. and L.H. provided beam transport simulations for the experiment design. D.P. designed the RPP. G.S. provided support in the target design. W.C., A.O., S.B., C.B., A.B., S.B., O.D., J.D., M.E., P.F., S.F., J.H., D.J., E.K., A.K., D.K., G.M., N.W.N., S.S., M.S., N.S., D.S., F.W., S.W. and Y.Z. were involved in the experiments. L.H., G.M., T.S. and M.R. conceived the project in this paper. W.C., J.V. and D.O.G. wrote the manuscript.

## Additional information

**Competing interests:** The authors declare no competing financial interests.

