## [Peer Review File · Nature Communications]

Reviewers' comments:

Reviewer #1 (Remarks to the Author):

Comments:

The Authors provide a report on the experimental test of ion stopping models near the Bragg peak in highly ionized matter. The energy loss of ions in dense plasmas is an important issue in ICF studies. Many theoretical models for the stopping power have been developed. By now, however, ion stopping in ionized plasmas is far from being understood because of various theoretical and experimental challenges. Only a few experimental data have been collected [for example, A. B. Zylstra et al., PRL 114, 215002 (2015), and J. A. Frenje et al., PRL 115, 205001 (2015)]. The Authors also performed a relevant experiment combined with detailedly theoretical analysis. The experimental design is interesting, and the results sound reasonable. The paper is generally well-written and the results are concisely presented.

I would like to ask the Authors to discuss the following points:

1. The present manuscript is not the first conclusive experimental test of stopping power models. For the first time, A. B. Zylstra et al. have quantitatively measured the ion stopping around the Bragg peak in high energy density plasmas [PRL 114, 215002 (2015)]. These experimental results represent the first sensitive tests of plasma stopping power theories around Bragg peak. If the Authors believe that their work is the first experimental test of stopping power models, please provide helpful and powerful evidences in the paper. If not, the title of the manuscript needs to be modified. Additionally, the Authors should discuss the differences between their work and the work of A. B. Zylstra et al. published in PRL 114, 215002 (2015).

2. In the present manuscript, the prediction of Li-Petrasso (LP) model is far from the experimental data. However, the experimental data in PRL 114, 215002 (2015) generally support the predictions of the Brown-Preston-Singleton (BPS) and LP stopping models. The Authors should provide clear reasons for this different conclusion.

3. The Authors have neglected the contributions from the plasma ions in their theoretical analysis because the contributions from plasma ions become significant for very low beam velocities, see the statements that "In most cases, we can neglect the contributions from plasma ions ... energy is deposited." in page 3 of the manuscript. In other words, the present experiment is a test of the ion stopping models in the one-component plasma (free electrons). However, in this manuscript, the Authors report the experimental data at low projectile velocities. I suggest the Authors provide numerical evidences that the contributions from the plasma ions can be neglected in the plasma parameter range they considered. For instance, the relative importance of ion and electron stopping calculated by using the LP model, the BPS model, and the T-matrix model should be clearly shown in the paper or a supplementary file.

4. The results show that the BPS predictions fit the experimental data quite well. It is known that the BPS model gives a complete description of both short-distance hard Coulomb collisions of the plasma particles and long-distance collective excitations of the plasma by the Boltzmann and the Lenard-Balescu equations. However, the Authors concluded that (page 9) a fully binary collision approach is required to understand their experimental data. This conclusion seems inconsistent with the BPS theory, please explain.

5. The BPS model contains three parts: (1) the classical short-distance contribution, (2) the classical

long-distance contribution, and (3) the quantum correction to the classical part. The classical BPS model may overestimate the energy loss of ion moving in the plasmas. The other models also include the quantum effect. So, corresponding to the present plasma conditions, the Authors need discuss the importance of quantum corrections in the stopping-power models.

6. The LP model includes the important effects of plasma ion stopping, collective plasma oscillations, and quantum effect. In the present manuscript, the Authors have neglected the effect of plasma ion stopping in their theoretical calculations. This may be the reason why the LP prediction is far from the experimental data. I suggest the Authors to reconsider the LP model via taking into account the effect of plasma ion stopping.

7. The Authors said that "the ion beam of originally $E_p=3.6$ MeV per nucleon energy is degraded through a $41\mu\text{m}$ thick carbon foil, resulting in a mean ion energy of $E_p=0.586\pm 0.016$ MeV per nucleon." It is not clear to me how to determine the degraded ion energy in experiments. The measured energy spectra of the degraded nitrogen ions should be additionally shown in the manuscript or a supplementary file.

To sum up, in my view, the experimental results obtained may be useful for testing the stopping-power models in plasmas, but not the first one. Furthermore, it just provides some experimental data in a weakly coupled plasma ($n_e=5*10^{20}$ cm^{-3} , $T_e=150$ eV, and $\Gamma=0.01$). And thus, from this experiment, one cannot conclude the validity of the theoretical models in much wider ranges of temperature and density of plasmas. Therefore, I cannot recommend the publication of this paper in Nature Communications.

Reviewer #2 (Remarks to the Author):

Review of manuscript NCOMMS-16-16897-T ("First conclusive experimental test of ion stopping models near the Bragg peak in highly ionized matter") by W Cayzac et al.

Obtaining a fundamental understanding of the ion-stopping power around the Bragg peak is an important and long-standing problem that many research groups have explored experimentally for the last few decades. However, and as stated by the authors, none of these groups have been able to differentiate various ion-stopping-power models around the Bragg peak because of poorly experimentally characterized plasma conditions. In this work, the authors claim that this has been achieved for the first time. Although I applaud that the authors seem to have done an excellent job experimentally going after this important but difficult problem, I am not sure about their claim because I do have concerns about the discussion and characterization of the projectile ions and target-plasma conditions, and the interpretation of the energy-loss data. I am not convinced that this work has adequately characterized the experiment conditions. I also feel that the conclusions rely too much on simulations. The specifics of my concerns are described below:

1. As the ion stopping scales to the first order with Z^2 , it is critical to have a good understanding of the charge state of the projectile ions. In this work, the charge state of the nitrogen ions has been determined through modeling, which is not entirely satisfying and a concern of mine. A discussion about the charge-state uncertainty has not been made as well. To address these issues, I think the authors should include experimental data that supports the modeling, and provide a better discussion about the projectile-charge state and its uncertainty.

2. The measured energy loss of nitrogen ions is mainly dictated by the areal density (ρR) and electron temperature (T_e) of the target plasma. Understanding these two parameters is therefore important for

assessing the ion stopping at the Bragg peak. Even though this the case, the authors only gloss over the characterization of these parameters in addition to the free electron density (n_e) by referring to refs. 11, 19 and 25. When reviewing these references and references therein, it is not clear to me how these parameters have been experimentally characterized and what the results are. This is not adequate and needs to be addressed. An additional concern is that the 2D simulated n_e only agree with measured on a 20% level. Could it be that the actual n_e in 3D is significantly lower than the 2D simulated?

3. It is not clear to the reader what the energy and velocity distributions of the projectile ions are after the energy degrader. How wide are these distributions and what is the average projectile velocity relative to the average velocity of the plasma electrons?

4. A simple calculation indicates that the electron-electron equilibration time is ~ 4 ns for these target-plasma conditions. That said, it is not clear to me that the electrons are in thermal equilibrium and that an electron temperature can be assigned to the target electrons. This should be elaborated upon.

In conclusion, before this manuscript can be considered for publication, a much better experimental characterization and discussion of the target-plasma conditions must be made.

Reviewer #3 (Remarks to the Author):

A. Summary of key results

The manuscript, "First conclusive experimental test of ion stopping models near the Bragg peak in highly ionized matter," by W. Cayzac et al, describes measurements of the energy loss of Nitrogen ions passing through an ideal carbon plasma in the vicinity of the Bragg peak. Energy loss is reported to be enhanced by up to 50% compared to that experienced in the cold matter case, with uncertainties tight enough to discriminate between different classes of stopping power theories.

B. Originality

The work reported here extends the earlier work in Frank et al (2013) [ref 11] to projectiles of lower energy such that the projectile speed is comparable to the thermal electron speed in the ideal plasma (i.e. the Bragg peak). This is an important regime in plasma stopping power theories, as it corresponds to an extremum of the stopping force, and generally is the regime where theories differ by the largest factor. Other previous work [references 13,14,15] have made measurements in ideal plasmas near the Bragg peak, but were unable to conclusively discriminate between the various models.

C. Validity of approach, quality of presentation

The basic experimental approach is sound, and the results rule out some theoretical models in this regime near the Bragg peak. However, the manuscript leaves out a number of details that are needed for a complete assessment of the reported data analysis and conclusions.

Firstly, none of the raw data is shown for either the energy loss or density measurements. It would be a nice illustration of the method to show how the sequence of data is obtained for the solid case, the plasma case, and the "vacuum" case.

Furthermore, the prescription to get between the input data and the digested results is incomplete. On

page 7, in particular, the text jumps back and forth between simulation results, calculated results, and experimental results. For example, the second to last paragraph starts with "The beam energy loss $[\Delta E]$ is calculated as the integral of the stopping power along the ion trajectory," and the following paragraph starts with "The energy loss was determined from the time shifts in the ion signals..." It is confusing which energy loss is used where, and this confusion is compounded by the fact that the energy losses are further "normalized by the value for the solid sample". This narrative needs to be clarified and polished.

On page 6, in paragraph 2, the free-electron density is claimed to be validated experimentally to better than 20%. Later, on page 8, it is claimed that the electron density has "been characterized experimentally to a high accuracy." However, the measurement uses laser interferometry, which likely does not measure the densest regions which dominate the energy loss. Density agreement in the blowoff plasma is not very surprising since there are very steep gradients. These claims would be better supported by showing a comparison of the measured and simulated density profiles.

D. Appropriate use of statistics and treatment of uncertainties

The authors are perhaps a bit inconsistent with the error ranges reported in the text. In some cases, the error range refers to the variation over different experiments (e.g. degrader thickness on page 10), and sometimes to the uncertainty of a particular value, but others are ambiguous as to the appropriate interpretation.

Nonetheless, the authors do offer uncertainty values for many of the critical parameters and measurements used in the analysis. However, these are scattered throughout the manuscript, and there is no explicit description of the dominant uncertainties, their relative contributions, or in particular how they propagate to the error bars shown in figs 2c and d.

The authors claim in the title to have a "conclusive experimental test", and in the conclusions that they can "unambiguously discriminate between different classes of stopping power theories", but no effort is made to quantify this discrimination.

E. Conclusions

Despite my criticisms on the clarity of the narrative and use of statistics, this is nonetheless an interesting and important result. I recommend the manuscript for publication after significant revision to improve the clarity of the data analysis procedure, as well as to include either an example of the raw data, or to give more detail regarding the intermediate steps.

F. Suggested improvements

In addition, I offer a number of smaller corrections and suggestions to the authors in an effort to improve the manuscript:

- p.3, paragraph 2: I recommend rewording the first sentence which contains the confusing sequence "self-heating via energy loss".
- p.3, paragraph 2: the phrase "...the Bragg peak where most of the ion energy is deposited" is imprecise. The Bragg peak corresponds to the maximum in the stopping power, but it is not necessarily the case where most of the ion energy is deposited in this peak, for example if the initial projectile energy is 10x the energy corresponding to the peak.

- p.4, paragraph 2: here it is claimed that the plasma target parameters are "similar to" the ones in an ICF plasma. This should be made quantitative either in this paragraph or in the following section. In the following section, typical density and temperature values, as well as the coupling and degeneracy parameters for the experiment are reported, but no quantitative comparison is made to typical burning or igniting ICF plasmas.
- p.5 fig1a: I recommend using consistent distance units for the two distances shown, i.e. 15 mm and 500 mm.
- p.5 last sentence: time "resolution" is reported as 0.25 ns, but in fact the setup cannot resolve peaks this closely spaced, due to the 5.5 ns duration of the ingoing nitrogen bunches and the 2.8 ns response time of the detector. What is probably meant here is the precision with which the peak center can be determined. The same concern applies to the reported "energy resolution" on the first line of p.6. This last instance is of particular concern due to the remainder of the sentence, which implies the ability to discriminate models at the 1% level, which is not the case presented in figure 2.
- p.6, paragraph 3: the simulated plasma areal density is said to agree with previous energy loss data, but how does it compare to the areal density at $t=0$? Namely, how much of the target areal density is eroded by plasma expansion? How does this vary with time?
- p.6, paragraph 3: There is a typo in the areal density equation (an extra factor of "x").
- p.6, paragraph 4: what relative contribution does the enhancement of Z_{eff} in the plasma have on the inferred energy loss and uncertainty?
- p.7, fig2b: indicate the uncertainty in the Z_{eff} calculation, either in the figure, or in the text.
- p.7, fig2d: a naive expectation is that the colder plasma energy loss curves should be closer to the cold-matter values, but this is the opposite of what is shown. This possible point of confusion is worth a brief description in the caption or the body text.
- p.8, paragraph 3: 20-25% is said to be "significantly above the experimental error bars." What are the displayed error bars? 1-sigma?
- p.8, paragraph 4: "This temperature range is deliberately taken much larger than realistic uncertainties in T_e ..." What is the realistic uncertainty, has that been quantified? Why not take that value instead of an unrealistic uncertainty?
- p.8, paragraph 4: The last sentence ends "...with a sufficient precision to disprove these theories." I recommend qualifying this statement with something like "in this parameter range", since these other theories still seem adequate in the high-velocity region for which they were originally designed, and indeed are stated to be appropriate in this regime on page 3 of this manuscript.
- p.8, last paragraph: the first sentence states these measurements are used to discriminate theories in "dense plasmas", and two sentences later the plasma conditions are instead described as "ideal and nondegenerate". These statements are confusing if not inconsistent. Based on the parameters reported in equation 2, the conditions seem to be weakly-coupled and non-degenerate, which are characteristics of low-density plasmas.
- p.9 "setup" paragraph 1: I recommend also stating the laser irradiance on target.

- p.10 "targets" paragraph 1: Does the initial areal density error range represent the range among different targets, or the precision of an individual target?

- There is significant temporal evolution of the plasma over the rather long (> 5 ns) arrival interval of the ion bunch. How this folds in to the analysis should be described explicitly (if briefly) in the body text. As it stands, there is only a passing remark in the caption to fig 2 regarding this step of the analysis.

G. References

References are adequate.

H. Clarity and context

Although much of the narrative needs significant revision to improve the clarity of the data analysis steps, the context given in the abstract, introduction, and conclusions are reasonably clear. A remark on page 9, "This finding has also strong implications for other transport and relaxation properties where similar electron-collisions play an important role," may be worth further elaboration, as these as-yet unidentified implications could help place stopping power in better context within the scope of Coulomb-mediated transport and relaxation properties in general.

Finally, I recommend the native english speaking co-authors review the text for proper articles and awkward phrasing. Examples include one in the first paragraph, "...allows to clearly rule out..." and in the last paragraph before the methods section "...in addition of their direct relevance...".

Detailed responses to each of the Referees' comments and suggestions

The Reviewers' comments are written in blue italic, while our responses are written in the standard style.

Reviewer #1

The Authors provide a report on the experimental test of ion stopping models near the Bragg peak in highly ionized matter. The energy loss of ions in dense plasmas is an important issue in ICF studies. Many theoretical models for the stopping power have been developed. By now, however, ion stopping in ionized plasmas is far from being understood because of various theoretical and experimental challenges. Only a few experimental data have been collected [for example, A. B. Zylstra et al., PRL 114, 215002 (2015), and J. A. Frenje et al., PRL 115, 205001 (2015)]. The Authors also performed a relevant experiment combined with detailedly theoretical analysis. The experimental design is interesting, and the results sound reasonable. The paper is generally well-written and the results are concisely presented.

I would like to ask the Authors to discuss the following points:

1. *The present manuscript is not the first conclusive experimental test of stopping power models. For the first time, A. B. Zylstra et al. have quantitatively measured the ion stopping around the Bragg peak in high energy density plasmas [PRL 114, 215002 (2015)]. These experimental results represent the first sensitive tests of plasma stopping power theories around Bragg peak. If the Authors believe that their work is the first experimental test of stopping power models, please provide helpful and powerful evidences in the paper. If not, the title of the manuscript needs to be modified. Additionally, the Authors should discuss the differences between their work and the work of A. B. Zylstra et al. published in PRL 114, 215002 (2015).*

We maintain our statement that our work provides the first conclusive experimental test of stopping-power theories near the Bragg peak. The Reviewer mentions the work by A.B. Zylstra et al. [PRL 114, 215002 (2015)] as the first sensitive test of this kind.

Firstly, the measurements by Zylstra et al. have been performed in the high-velocity stopping regime ($v_p/v_{th}^e > 10$), where the stopping power reaches the well-known Bethe limit. The specificity of their work lies in the fact that the plasma is partially ionized, hence the relative contributions from free and bound electrons to the stopping power can be determined, as well as mean target ionization potentials, knowing the stopping-power formalism. However, theories describing stopping by free electrons all agree in this parameter regime and they cannot be discriminated.

Secondly, recently there has also been the work by J.A. Frenje et al. [PRL 115, 205001 (2015)], which reports experimental data near the Bragg peak in a highly-ionized plasma and compares them with the LP and BPS models, similarly to our work. However, although the work by Frenje et al. presents quantitative stopping measurements near the Bragg peak and studies their dependence on the plasma parameters, their data do not allow to favour one stopping-power model over another, as is explicitly written at the end of their publication:

"The BPS and LP formalisms, with 25%–30% quantum reduction to the ion stopping, agree with the data for $v_i > v_{ih}$. There are some differences at $v_i \sim v_{ih}$, but the current data set cannot distinguish between them".

In contrast, our data distinguish between the BPS and the LP models. For the first time, we show that the "standard" LP formalism is inaccurate near the Bragg peak as it overevaluates the stopping power, while the "more advanced" BPS (as well as TM) models reproduce our data correctly.

Therefore, our data are conclusive, showing that only a theory that includes a detailed description of close collisions is accurate in this parameter range. Still, we have taken the Referee's criticism into account and we changed the title of the paper to "*Experimental discrimination of ion stopping models near the Bragg peak in highly ionized matter*", removing the mention "first". Additionally, we changed the word "test" into "discrimination", which is more precise.

Why our work distinguishes between both formalisms

Our work distinguishes between both formalisms because the considered beam-plasma configuration leads to significantly larger energy-loss differences between the stopping-power models.

Firstly, the stopping power and, thus, also the differences between the models, are significantly larger in our work. This is due to the fact that, on the one hand, we use heavier ions (nitrogen instead of protons or alpha particles), which feature a much higher charge state. And on the other hand, our plasma features a significantly lower temperature (typically 150 eV instead of typically a few keV in the work by Frenje et al.). As the stopping power is a decreasing function of the temperature in the whole range of interest, this also contributes to the much larger stopping power in the conditions of our work. In typical conditions investigated by Frenje et al., the stopping power is of the order of 70-80 keV/(mg/cm²), as shown in Fig. 2 in their paper. In our case, the stopping power is of 15-20 keV/(μg/cm²), i.e. by a factor 200-300 larger.

Secondly, the fact that we use heavier ions leads to a larger beam-plasma coupling. Indeed, the Coulomb parameter for the beam-plasma interaction is $\eta = Z_b e^2 / (\hbar v_r) \approx 1$, Z_b being the beam charge and v_r the relative velocity between the projectile ions and the plasma electrons, as we now indicate on page 5 of the manuscript. This means that the contribution of close binary, nonperturbative collisions to the stopping power is significant. As such collisions are more accurately described by the BPS (or TM) formalism, the differences between the predictions of the perturbative LP and of the non-perturbative BPS stopping models are further enhanced.

As a result, the differences in stopping power between the LP and the BPS theories are much more important for our beam-plasma conditions than in those of Frenje et al., making them easier to access experimentally.

To address and clarify these points in the manuscript, we reformulated the results analysis and the conclusion in the main text to a large extent (page 10).

2. In the present manuscript, the prediction of Li-Petrasso (LP) model is far from the experimental data. However, the experimental data in PRL 114, 215002 (2015) generally support the predictions of the Brown-Preston-Singleton (BPS) and LP stopping models. The Authors should provide clear reasons for this different conclusion.

The beam-plasma configuration in our work leads to discrepancies in energy loss resulting from the different stopping-power formalisms by a factor 3-6 larger than in the work by Frenje et al., for the reasons explained in point 1. This makes it possible to distinguish between these theories experimentally, as the gap between the predictions is larger than the experimental error bars.

3. The Authors have neglected the contributions from the plasma ions in their theoretical analysis because the contributions from plasma ions become significant for very low beam velocities, see the statements that "In most cases, we can neglect the contributions from plasma ions ... energy is deposited." in page 3 of the manuscript. In other words, the present experiment is a test of the ion stopping models in the one-component plasma (free electrons). However, in this manuscript, the Authors report the experimental data at low projectile velocities. I suggest the Authors provide numerical evidences that the contributions from the plasma ions can be neglected in the plasma

parameter range they considered. For instance, the relative importance of ion and electron stopping calculated by using the LP model, the BPS model, and the T-matrix model should be clearly shown in the paper or a supplementary file.

Indeed, we report experimental data at "low projectile velocities", but the velocity in our experiment is still considerably larger than the thermal velocity of the plasma ions (we have $v_p/v_{th}^i \sim 175$, while $v_p/v_{th}^e \sim 1$, where v_{th}^e and v_{th}^i are the electron and ion thermal velocities, respectively). The contribution of plasma ions would only play a role for a projectile energy close to 0.02 keV per nucleon. Therefore the projectile velocity in our experiment is very far above this velocity range, and there is no need of checking this in more detail. To make it clear in the manuscript, we modified the introduction (page 3) to define both the electronic and the ionic thermal velocity, and on page 5 after mentioning the typical values of the plasma parameters and the beam energy, we indicated the corresponding values for v_p/v_{th}^e and v_p/v_{th}^i in order to justify that we neglect the ion contribution.

4. The results show that the BPS predictions fit the experimental data quite well. It is known that the BPS model gives a complete description of both short-distance hard Coulomb collisions of the plasma particles and long-distance collective excitations of the plasma by the Boltzmann and the Lenard-Balescu equations. However, the Authors concluded that (page 9) a fully binary collision approach is required to understand their experimental data. This conclusion seems inconsistent with the BPS theory, please explain.

Here, our sentence "instead, a full binary collision approach is required" was indeed misleading. We have tried to say that a correct stopping-power approach needs to *include* a full description of binary collisions, in addition to the small-angle collisions and collective effects that are already accounted for by the perturbative stopping models. Therefore we replaced the sentence with "Instead, approaches that *include* a full description of binary collisions agree with the data." (top of page 11).

5. The BPS model contains three parts: (1) the classical short-distance contribution, (2) the classical long-distance contribution, and (3) the quantum correction to the classical part. The classical BPS model may overestimate the energy loss of ion moving in the plasmas. The other models also include the quantum effect. So, corresponding to the present plasma conditions, the Authors need discuss the importance of quantum corrections in the stopping-power models.

There are two distinct ways in which quantum effects can influence the stopping power. The first one is due to the statistics of a many-particle system, namely Fermi statistics in degenerate systems. This would for instance restrict possible final states in the scattering process but is not important in our case since the plasma is nondegenerate.

The second class of quantum effects is due to quantum diffraction. It is important for any scattering process where the De Broglie wavelength of the scattering electron is of the same order as the interaction zone (often estimated by the screening length). In many beam-plasma configurations, this is the case especially when describing the interaction with the fast electrons in the tail of the distribution. Hence, these effects also play a role in hot plasmas and a proper theory for the collision processes, and thus for the stopping power, should be based on quantum mechanics. In fact, only using quantum mechanics is it possible to correctly describe a close binary collision of electrons and ions. In such treatment, no Coulomb divergence occurs as it is known from classical scattering theory of Rutherford or classical plasma theory of Spitzer or Landau and, thus, there are no free parameters (Coulomb logarithm). Therefore quantum diffraction effects are essential to correctly describe binary collisions.

Both the T-matrix and the BPS theories include quantum diffraction effects in addition to the full non-perturbative nature of the collisions, while the LP scheme is mainly based on classical

collisions. Part of the deviations found in our comparison might thus be attributed to the inadequate description of collisions for hot plasmas within the LP formalism.

In the manuscript, we now estimate the importance of quantum diffraction effects using the Coulomb parameter (page 5), in the sentence "Moreover, the Coulomb parameter $\eta = Z_b e^2 / (\hbar v_r)$, Z_b being the beam charge and v_r the relative velocity between the projectile ions and the plasma electrons, is roughly unity. This indicates a significant beam-plasma coupling and thus the importance of strong non-perturbative collisions, while quantum diffraction effects still play a role." Moreover, we have modified the following sentence on page 8 when mentioning the T-matrix and BPS theories: "Furthermore, we compare to the model by Brown-Preston-Singleton (BPS) and the T-matrix formulation employing a velocity-dependent screening length (TM), which both include a detailed treatment of close binary collisions *as well as quantum diffraction effects*".

6. The LP model includes the important effects of plasma ion stopping, collective plasma oscillations, and quantum effect. In the present manuscript, the Authors have neglected the effect of plasma ion stopping in their theoretical calculations. This may be the reason why the LP prediction is far from the experimental data. I suggest the Authors to reconsider the LP model via taking into account the effect of plasma ion stopping.

As explained in point 3, there is no contribution from the plasma ions in our conditions, hence the stopping power originates exclusively from the plasma electrons. The LP prediction is far from the data because it is essentially perturbative and includes the effects of close binary collisions only as a second-order correction and because collisions are described mainly classically as mentioned in point 5. Theories that fit the data well take close collisions into account accurately because they are based on a fully quantum mechanical description of collisions (that is, solutions of the Schrodinger equation for the T-matrix approach).

7. The Authors said that "the ion beam of originally $E_p=3.6$ MeV per nucleon energy is degraded through a $41\mu\text{m}$ thick carbon foil, resulting in a mean ion energy of $E_p=0.586\pm 0.016$ MeV per nucleon." It is not clear to me how to determine the degraded ion energy in experiments. The measured energy spectra of the degraded nitrogen ions should be additionally shown in the manuscript or a supplementary file.

In order to detail and illustrate the analysis procedure of the energy-loss data, we added a sample of raw detector data from one of the shots in the methods (Fig. 4 on page 15). Here, we show signals through the solid target and through the plasma. We also detailed the explanations in the text of the "Data analysis" section (pages 14-16). In particular, it should now be clear to the reader how the mean degraded ion energy is determined in the experiments. First, the ion energy after the filter is deduced from the time-of-flight measurement. Second, a SRIM simulation is performed to trace back to the ion energy after the degrader, knowing the thickness of the target and of the filter precisely. In this way, the mean degraded ion energy is determined with a precision of 0.005 to 0.010 MeV per nucleon.

Conclusion

*To sum up, in my view, the experimental results obtained may be useful for testing the stopping-power models in plasmas, but not the first one. Furthermore, it just provides some experimental data in a weakly coupled plasma ($n_e=5*10^{20} \text{ cm}^{-3}$, $T_e=150 \text{ eV}$, and $\Gamma=0.01$). And thus, from this experiment, one cannot conclude the validity of the theoretical models in much wider ranges of temperature and density of plasmas. Therefore, I cannot recommend the publication of this paper in Nature Communications.*

We thank Referee #1 for stressing the relevance and the quality of our work, and for his constructive criticism as well as his suggestions to improve the theoretical discussion, in particular to better justify the specificity of our work in comparison with previous publications.

In conclusion, we maintain that our results are the first of their kind, enabling to distinguish between two classes of stopping-power theories and hence of kinetic descriptions of particle transport in plasma for the important parameter range of the Bragg peak.

We also stress the fact that even though the presented data were obtained in weakly-coupled plasma, the plasma coupling parameter Γ and the degeneracy parameter Θ are similar to the ones of a burning ICF plasma ($\Gamma \sim 0.01$ and $\Theta \sim 550$ in our case versus $\Gamma \sim 0.01$ and $\Theta \sim 30$ for typical parameters of $n_e = 10^{25}/\text{cm}^3$ and $T_e = 5000$ eV, as mentioned on top of page 5). In both cases, the plasma is nondegenerate with a similar plasma coupling strength, which means that the plasma regime is essentially the same. This fact emphasizes the relevance of our work for hot fusion plasmas. Moreover, our results show that even for weakly coupled and nondegenerate plasmas, there is a significant velocity effect on the stopping power around the Bragg peak due to close collisions between the projectile ions and the plasma electrons, which is a new significant finding in itself.

Reviewer #2

Obtaining a fundamental understanding of the ion-stopping power around the Bragg peak is an important and long-standing problem that many research groups have explored experimentally for the last few decades. However, and as stated by the authors, none of these groups have been able to differentiate various ion-stopping-power models around the Bragg peak because of poorly experimentally characterized plasma conditions. In this work, the authors claim that this has been achieved for the first time. Although I applaud that the authors seem to have done an excellent job experimentally going after this important but difficult problem, I am not sure about their claim because I do have concerns about the discussion and characterization of the projectile ions and target-plasma conditions, and the interpretation of the energy-loss data. I am not convinced that this work has adequately characterized the experiment conditions. I also feel that the conclusions rely too much on simulations.

The specifics of my concerns are described below:

1. *As the ion stopping scales to the first order with Z^2 , it is critical to have a good understanding of the charge state of the projectile ions. In this work, the charge state of the nitrogen ions has been determined through modeling, which is not entirely satisfying and a concern of mine. A discussion about the charge-state uncertainty has not been made as well. To address these issues, I think the authors should include experimental data that supports the modeling, and provide a better discussion about the projectile-charge state and its uncertainty.*

The treatment of the projectile charge state has been improved in two aspects. Firstly, we use two additional charge-state models in order to show the deviations between the various predictions and, thus, to evaluate the uncertainty in the charge-state modeling (Fig. 3a), and we show the effect of the three charge-state models on the energy-loss calculation (Fig. 3b). Secondly, we add experimental data that support the modeling for a case of high projectile velocity where the uncertainty on the stopping power is reduced (Fig. 2d).

In Fig. 2d, we show the experimental energy-loss data from the work by Frank et al. that were

obtained at the same plasma conditions, for argon projectiles and for a higher velocity ratio ($v_p/v_{th}^e \sim 3$). These data are compared with the LP and TM stopping-power formalisms as in our results panel (Fig. 3), considering the models by Gus'kov et al. and by Kreussler et al. for the effective beam charge state as well as a Monte-Carlo (MC) model based on the cross sections of charge-exchange processes. The first two models are based on a similar formalism and give similar results, whereas the MC model predicts a significantly higher charge state. Note that, strictly speaking, the Gus'kov model predicts an effective charge state, while the Kreussler and MC models calculate a mean charge state, but for relatively light ions like nitrogen there is virtually no difference between the two quantities (as we discuss in the methods).

For the case of Fig. 2d, the differences between the stopping-power models and between the beam charge-state models remain small and the experimental data do not enable to distinguish between them. Therefore, any of the charge-state and stopping-power models tested leads to a good agreement between the simulated energy loss and the experimental data.

In the beam-plasma configuration of our work, the gap between the charge-state models increases (Fig. 3a), which also causes larger differences between the energy-loss results calculated applying these three models (Fig. 3b). It is shown that our data can only be fitted by the energy-loss simulation using either the Gus'kov or the Kreussler charge model combined with the T-matrix description for the stopping power. This shows that the LP stopping-power model, as well as other standard perturbative models, are disproved for any charge-state modeling. On the other hand, the experimental data support the Gus'kov and the Kreussler charge-state formalisms.

2. The measured energy loss of nitrogen ions is mainly dictated by the areal density (ρR) and electron temperature (T_e) of the target plasma. Understanding these two parameters is therefore important for assessing the ion stopping at the Bragg peak. Even though this the case, the authors only gloss over the characterization of these parameters in addition to the free electron density (n_e) by referring to refs. 11, 19 and 25. When reviewing these references and references therein, it is not clear to me how these parameters have been experimentally characterized and what the results are. This is not adequate and needs to be addressed. An additional concern is that the 2D simulated n_e only agree with measured on a 20% level. Could it be that the actual n_e in 3D is significantly lower than the 2D simulated?

In this revised version of the manuscript, we improved the analysis and the discussion on the characterization of the plasma parameters. In particular, experimental results for the benchmarking of the plasma areal density and the free electron density are presented in Fig. 2. The temperature is deduced from the simulations. The uncertainties in the values of the density and temperature are taken into account in the error analysis of the simulated energy loss.

Areal density

The areal density is benchmarked against the experimental energy-loss data from the work by Frank et al. as shown in Fig. 2d. These measurements were obtained for the same plasma conditions, for argon projectiles and for a higher velocity ratio ($v_p/v_{th}^e \sim 3$). Stopping-power calculations show that even if the stopping power strongly depends on the temperature at the Bragg peak ($v_p/v_{th}^e \sim 1$), it is virtually independent from the temperature for $v_p/v_{th}^e \sim 3$, and it only slowly varies with the free electron density. Moreover, all stopping-power models agree for this relatively large v_p/v_{th}^e ratio. Hence the energy loss essentially depends only on the areal density of the plasma, whose temporal evolution is also displayed in Fig. 2d. The good agreement between all our predictions and the experimental data (which do not enable to distinguish between the models) proves that the temporal evolution of the areal density is known with a high precision, even at times $t = 13-15$ ns where it has decreased by 50% due to three-dimensional plasma expansion.

Free electron density

In Fig. 2b we present two examples of results from laser interferometry measurements, showing experimentally determined free electron density profiles for $t = 7$ and 11 ns (before and after the start of the 3D plasma expansion, respectively). These data are compared with the corresponding profiles extracted from the plasma simulations, similarly to the graphs shown in the works by Frank et al. and Boerner et al. Our measurements, analyzed over a time range from $t = 5$ to 15 ns, reveal maximum differences between the measured and simulated profiles by a factor of roughly 2. Still, in most cases, these differences are much smaller (of around 20%). The latter statement is also supported by the fact that the simulated areal density, which is the integral of the density over the plasma length, is in good agreement with the energy-loss measurements of Fig. 2d. The fact that the areal density is precisely known also suggests that the actual 3D density profile cannot be very different from the 2D simulated density profile. Therefore, the factor 2 on the uncertainty in the free electron density constitutes an upper boundary of the largest possible error.

The effect of the uncertainty in the free electron density on the simulated energy loss is discussed on page 10 and illustrated on Fig. 3d. We find that the factor 2 uncertainty in the free electron density leads to an uncertainty of up to 10% in the simulated energy loss, which is significant but remains smaller than differences between theories and, thus, does not question the validity of our conclusions.

Electron temperature

Complementary simulations indicate that the maximum uncertainty in the free electron density by a factor 2 implies a maximum uncertainty of ± 40 eV in the temperature. The effect of this uncertainty on the energy loss simulation is also discussed on page 10 and included in Fig. 3d, leading to an error of up to 10-15% on the simulated energy loss. Hence, the combined error on the energy loss due to the uncertainties in density and temperature reaches a maximum value of 15%, which is smaller than the gap from the predictions of the standard perturbative stopping-power models to the experimental data.

3. *It is not clear to the reader what the energy and velocity distributions of the projectile ions are after the energy degrader. How wide are these distributions and what is the average projectile velocity relative to the average velocity of the plasma electrons?*

In the main text, we mention the measured mean energy (in the range 0.586 ± 0.016 MeV per nucleon), as well as the corresponding velocity ratio $v_p/v_{th}^e = 1.2$ (page 5). We indicate the width of the energy and of the velocity distributions of the beam after the degrader in the methods (page 12), as simulated with the TRIM code, namely: 5% in energy at 1σ (i.e. ± 30 keV per nucleon), which corresponds to less than 3% in velocity. Calculations show that the influence of this energy interval on the simulated energy loss is of the order of 1%, which can be neglected compared with the uncertainties in the plasma parameters.

4. *A simple calculation indicates that the electron-electron equilibration time is ~ 4 ns for these target-plasma conditions. That said, it is not clear to me that the electrons are in thermal equilibrium and that an electron temperature can be assigned to the target electrons. This should be elaborated upon.*

The electron equilibration time does not constitute an issue for the plasma thermal equilibrium, as is explained in the "Simulations" paragraph of the "Methods" section (page 17). We mention that the energy deposited in the plasma by one ion bunch is, by far, negligible compared to the global thermal energy of the plasma. Therefore the projectile energy loss in the plasma has no influence on the plasma properties and dynamics.

Conclusion

In conclusion, before this manuscript can be considered for publication, a much better experimental characterization and discussion of the target-plasma conditions must be made.

We want to thank Referee 2 for his praise on the achievement of our work as well as for his useful comments on the adequate characterization of the experiment conditions. We agree that a more detailed discussion about the projectile charge state as well as about the characterization of the plasma temperature and areal density considerably improved our results for their publication.

Although the beam and the plasma conditions have not been characterized with a very high precision, we systematically quantify the effects of all uncertainties on the simulated energy loss and we clearly demonstrate that our experimental energy-loss data still enable to distinguish the LP from the TM (or BPS) stopping-power models. This result remains valid for any of the used charge-state models, the Gus'kov model providing the closest fit to the measurements.

Therefore, our beam-plasma configuration is characterized *with a sufficient precision* to allow a proper discrimination of stopping-power theories, due to the large differences between the predictions of the different theories for the conditions investigated.

Reviewer #3

A. Summary of key results

The manuscript, "First conclusive experimental test of ion stopping models near the Bragg peak in highly ionized matter," by W. Cayzac et al, describes measurements of the energy loss of Nitrogen ions passing through an ideal carbon plasma in the vicinity of the Bragg peak. Energy loss is reported to be enhanced by up to 50% compared to that experienced in the cold matter case, with uncertainties tight enough to discriminate between different classes of stopping power theories.

B. Originality

The work reported here extends the earlier work in Frank et al (2013) [ref 11] to projectiles of lower energy such that the projectile speed is comparable to the thermal electron speed in the ideal plasma (i.e. the Bragg peak). This is an important regime in plasma stopping power theories, as it corresponds to an extremum of the stopping force, and generally is the regime where theories differ by the largest factor. Other previous work [references 13,14,15] have made measurements in ideal plasmas near the Bragg peak, but were unable to conclusively discriminate between the various models.

C. Validity of approach, quality of presentation

The basic experimental approach is sound, and the results rule out some theoretical models in this regime near the Bragg peak. However, the manuscript leaves out a number of details that are needed for a complete assessment of the reported data analysis and conclusions.

Firstly, none of the raw data is shown for either the energy loss or density measurements. It would be a nice illustration of the method to show how the sequence of data is obtained for the solid case, the plasma case, and the "vacuum" case.

We now show raw data for the energy-loss as well as the density measurements.

In Fig. 2, we show:

- in Fig. 2a, one example of raw interferometry data (measurement for a time of 11 ns after the start of the plasma heating as well as the corresponding reference measurement).
- in Fig. 2b, two examples for the experimentally obtained free electron density profiles, for times $t = 7$ and 11 ns, compared with the simulated profiles.

In Fig. 4 in the methods, we show a sample of the raw detector data for the energy-loss analysis from one laser shot with measured ion bunch signals in the solid and in the plasma. Signals in the "vacuum" case cannot be shown on the same picture but their period, which serves as "undisturbed" reference for the measurement, is represented by the vertical dotted lines. The data analysis process is also explained more in detail and the uncertainty analysis is clarified, with a description of the various error contributions.

Furthermore, the prescription to get between the input data and the digested results is incomplete. On page 7, in particular, the text jumps back and forth between simulation results, calculated results, and experimental results. For example, the second to last paragraph starts with "The beam energy loss $[\Delta E]$ is calculated as the integral of the stopping power along the ion trajectory," and the following paragraph starts with "The energy loss was determined from the time shifts in the ion signals..." It is confusing which energy loss is used where, and this confusion is compounded by the fact that the energy losses are further "normalized by the value for the solid sample". This narrative needs to be clarified and polished.

We modified the narrative substantially throughout the manuscript, not only in the part on the presentation of the results. In particular, we paid special attention not to generate any confusion between experimental results and simulation results and we notably used the nomenclature ΔE_{exp} and ΔE_{sim} for clarity.

On page 6, in paragraph 2, the free-electron density is claimed to be validated experimentally to better than 20%. Later, on page 8, it is claimed that the electron density has "been characterized experimentally to a high accuracy." However, the measurement uses laser interferometry, which likely does not measure the densest regions which dominate the energy loss. Density agreement in the blowoff plasma is not very surprising since there are very steep gradients. These claims would be better supported by showing a comparison of the measured and simulated density profiles.

We show a comparison between the measured and the simulated free electron density profiles in Fig. 2b for two different times $t = 7$ and 11 ns (before and after 3D plasma expansion has started). After a more complete data analysis over the whole temporal range of the measurements, we find maximum differences between the measured and the simulated free electron density profiles by a factor of two. In a majority of cases the agreement is significantly better, of around 20%, which is the value we reported in the first manuscript version. Therefore, we consider a maximum uncertainty by a factor of two as an upper boundary of the possible error. Even if this constitutes a much larger uncertainty than stated before, we show in our error analysis on the energy loss, that it does not prevent the discrimination of the stopping-power theories.

The Referee is right that the laser interferometry is not able to measure the density in the densest plasma regions where the energy loss is largest. However, the areal density has been benchmarked against energy-loss measurements (see Fig. 2d) and it corresponds to the integral of the density along the ion propagation through the plasma. Therefore the plasma density and thus the free electron density in the plasma centre cannot deviate significantly from the simulation predictions.

The fact that the areal density is known also implies that the uncertainty by a factor 2 deduced from the benchmarking of the free electron density profiles is a conservatively estimated maximum value over the whole spatial profile, but that in average, the density is known to a much better precision (mostly in the range of 20%).

The error on the energy loss originating from the factor 2 uncertainty in the free electron density is discussed on page 10 and turns out to be at most 10%, which is less than the error of up to 15% due to the uncertainty in the temperature, estimated to be ± 40 eV by complementary simulations based on the density benchmarking.

D. Appropriate use of statistics and treatment of uncertainties

The authors are perhaps a bit inconsistent with the error ranges reported in the text. In some cases, the error range refers to the variation over different experiments (e.g. degrader thickness on page 10), and sometimes to the uncertainty of a particular value, but others are ambiguous as to the appropriate interpretation.

Nonetheless, the authors do offer uncertainty values for many of the critical parameters and measurements used in the analysis. However, these are scattered throughout the manuscript, and there is no explicit description of the dominant uncertainties, their relative contributions, or in particular how they propagate to the error bars shown in figs 2c and d.

The Referee is right that the treatment of uncertainties was incomplete and sometimes confusing. In order to address this important aspect, we clarified the description of the uncertainties and we detailed all uncertainty values that are relevant in this work for the discrimination of the stopping-power theories.

Experimental uncertainties

The sources of uncertainty as well as their respective contributions are detailed in the section "Data analysis" in the methods (pages 14-16). A more detailed and more realistic error analysis has been performed. In particular, the error due to the uncertainty in the initial areal density of the target has been corrected as this areal density is known with a precision of 1%. On the other hand, systematic errors from the data analysis process have now been re-evaluated. As a result, the experimental error bars of Fig. 3.b-c-d have been modified. However, the newly determined values are similar to the previous ones and do not change the data interpretation and the work conclusions.

Uncertainties in the beam-plasma conditions

We completed the analysis of the uncertainties in the simulations, originating from:

- the energy distribution of the beam (5% at 1σ)
- the uncertainty in the beam charge state modeling (deviations between the calculations applying the different models)
- the plasma electron density (maximum by a factor 2 according to the simulation benchmarking against the results from the interferometry measurements)
- the plasma temperature (maximum by ± 40 eV according to complementary simulations)

All contributions are quantified and discussed in the text, and their influence on the energy loss is evaluated. The uncertainty originating from the energy distribution of the beam is much smaller than the other contributions and can thus be neglected (see page 12). The uncertainty in the beam charge state is evaluated with the deviations between the predictions of the different models as shown in Fig. 3.a-b. As the Gus'kov model appears to be most accurate to describe the experimental energy-loss data, it is the one used in the following in Fig. 3.c-d. The global error on the simulated

energy loss due to the uncertainties in the plasma density and temperature is illustrated in Fig. 3d in a final comparison between the energy-loss simulations and measurements.

The authors claim in the title to have a "conclusive experimental test", and in the conclusions that they can "unambiguously discriminate between different classes of stopping power theories", but no effort is made to quantify this discrimination.

The discrimination of the different classes of stopping-power theories is now clearly established by means of the error analysis illustrated in Fig. 3d. Despite the uncertainties in the plasma parameters and in the beam charge state as well as the size of the experimental error bars, we demonstrate that the LP model is disproved, while the TM and the BPS stopping models are supported, at the same time as the Gus'kov and the Kreussler charge-state formalisms.

As the displayed experimental error bars correspond to one standard deviation (1σ) of the measured energy loss, and the uncertainties on the simulations correspond to 2σ , we thus claim that we have discriminated the two different classes of stopping-power theories at least at 1σ . It is mentioned on page 10 in the manuscript in the sentence "In conclusion, our data provide a first conclusive test of stopping-power predictions for the velocity range around the Bragg peak by discriminating between different classes of theories in highly ionized plasmas to better than 1σ ".

E. Conclusions

Despite my criticisms on the clarity of the narrative and use of statistics, this is nonetheless an interesting and important result. I recommend the manuscript for publication after significant revision to improve the clarity of the data analysis procedure, as well as to include either an example of the raw data, or to give more detail regarding the intermediate steps.

F. Suggested improvements

In addition, I offer a number of smaller corrections and suggestions to the authors in an effort to improve the manuscript:

- p.3, paragraph 2: I recommend rewording the first sentence which contains the confusing sequence "self-heating via energy loss".

We modified the sentence to "In inertial confinement fusion (ICF), the target *self-heating due to the energy loss* of the 3.5 MeV α -particles born from deuterium-tritium (DT) fusion reactions must dominate all loss processes" (page 3).

- p.3, paragraph 2: the phrase "...the Bragg peak where most of the ion energy is deposited" is imprecise. The Bragg peak corresponds to the maximum in the stopping power, but it is not necessarily the case where most of the ion energy is deposited in this peak, for example if the initial projectile energy is 10x the energy corresponding to the peak.

We modified the phrase to "...a considerable part of the ion energy is deposited..." (page 3, after also changing a few other previous sentences).

- p.4, paragraph 2: here it is claimed that the plasma target parameters are "similar to" the ones in an ICF plasma. This should be made quantitative either in this paragraph or in the following section. In the following section, typical density and temperature values, as well as the coupling and degeneracy parameters for the experiment are reported, but no quantitative comparison is

made to typical burning or igniting ICF plasmas.

The claim that the plasma parameters are similar to the ones of a burning ICF plasma was quantified by adding the sentence "These conditions are relevant to burning ICF plasmas ($\Gamma \sim 0.01$ and $\Theta \sim 30$ for $n_e = 10^{25}/\text{cm}^3$ and $T_e = 5000 \text{ eV}$)" after giving the typical Γ and Θ values for our experiment (page 5).

- p.5 fig1a: I recommend using consistent distance units for the two distances shown, i.e. 15 mm and 500 mm.

All the distance units of Fig. 1 were converted in mm. The exact experimental time-of-flight distance of 462 mm is now indicated instead of the previously mentioned 500 mm value.

- p.5 last sentence: time "resolution" is reported as 0.25 ns, but in fact the setup cannot resolve peaks this closely spaced, due to the 5.5 ns duration of the ingoing nitrogen bunches and the 2.8 ns response time of the detector. What is probably meant here is the precision with which the peak center can be determined. The same concern applies to the reported "energy resolution" on the first line of p.6. This last instance is of particular concern due to the remainder of the sentence, which implies the ability to discriminate models at the 1% level, which is not the case presented in figure 2.

The time "resolution" is indeed the optimum precision with which the center of gravity of the ion peaks can be determined. It can be viewed as the "intrinsic" time resolution of the setup, which would be reached if there were no additional experimental uncertainties. The same applies to the "energy resolution".

We modified the sentence in the main text (page 5) to "The energy loss of the beam ions is measured using the time-of-flight (TOF) method, with a semi-conductor detector based on chemical vapour deposition (CVD) diamond (see Fig.1c). Our detector permits an *intrinsic* energy resolution of $\Delta E_p \approx 70 \text{ keV}$ ($\Delta E_p/E_p \approx 1\%$), which is much smaller than the differences between the predictions of several stopping-power models". Note that we corrected the 50 keV to the more accurate value of 70 keV for the parameters of our experiment. As for the time resolution of the detector, we now mention it only in the methods in the section "TOF diagnostic", in the sentence "The intrinsic detector resolution of 0.25 ns, i.e. the maximum precision on the determination of the centre of gravity of the ion bunch signals, combined with a 462 mm TOF distance, implies an intrinsic energy resolution of $\Delta E_p \approx 70 \text{ keV}$." (page 13).

At the end of the "Data analysis" section in the methods (page 16), we also added two sentences to echo back to the above sentences, namely: "Hence, the experimental error on the energy loss is significantly larger than the intrinsic energy resolution of the setup of 0.07 MeV and does not allow a fine benchmarking of theories. However, it still enables to clearly discriminate two classes of stopping-power theories and, in particular, to disprove the perturbative models." This makes clear that, due to the mentioned uncertainties, the measurement precision is not as good as it could be (we cannot resolve 1% of the initial energy, but rather 2-4%), but this is still sufficient to distinguish between the "standard" and the "advanced" theories.

- p.6, paragraph 3: the simulated plasma areal density is said to agree with previous energy loss data, but how does it compare to the areal density at t=0? Namely, how much of the target areal density is eroded by plasma expansion? How does this vary with time?

The simulated temporal evolution of the plasma areal density is presented in Fig. 2d and it is commented in the main text on pages 5-6. The same areal density curve is further displayed on the

energy-loss result graphs of Fig. 3b-c-d as an indication of the 3D target expansion.

- p.6, paragraph 3: *There is a typo in the areal density equation (an extra factor of "x").*

The typo was corrected in the areal density equation as well as in equation (3) (pages 5 and 8).

- p.6, paragraph 4: *what relative contribution does the enhancement of Z_{eff} in the plasma have on the inferred energy loss and uncertainty?*

We now use three different models for the effective charge state of the ion beam: the Gus'kov and the Kreussler models, as well as a Monte-Carlo model based on the cross sections of charge-exchange processes. The differences between those models are shown in Fig. 3a and the resulting differences on the calculated energy loss are presented in Fig. 3b. When using the Gus'kov or the Kreussler model, the charge-state contribution to the energy-loss enhancement in plasma is of up to 19%, while it reaches 45% using the MC model, as is mentioned in the text on pages 7-8.

- p.7, fig2b: *indicate the uncertainty in the Z_{eff} calculation, either in the figure, or in the text.*

The uncertainty in the effective charge-state modeling is estimated by the deviations between the energy-loss calculations applying the three considered models. The Gus'kov model, that predicts the lowest charge-state values (very similar to the ones predicted by the Kreussler model) appears as the most accurate to describe our experimental data and it is thus the one considered in the further analysis (Fig. 3c-d).

- p.7, fig2d: *a naive expectation is that the colder plasma energy loss curves should be closer to the cold-matter values, but this is the opposite of what is shown. This possible point of confusion is worth a brief description in the caption or the body text.*

The sentence "Note that the stopping power decreases with temperature and with density in the considered parameter range." was added on page 10, as we also evaluate the sensitivity of the energy-loss calculation to the free electron density.

- p.8, paragraph 3: *20-25% is said to be "significantly above the experimental error bars." What are the displayed error bars? 1-sigma?*

The displayed error bars are 1σ , and this has now been indicated on page 8: "The error bars correspond to one standard deviation (1σ) of the uncertainty in the time shifts".

- p.8, paragraph 4: *"This temperature range is deliberately taken much larger than realistic uncertainties in T_e ..." What is the realistic uncertainty, has that been quantified? Why not take that value instead of an unrealistic uncertainty?*

The uncertainty in the temperature has been quantified with the help of complementary hydrodynamic simulations, using the benchmarking of the simulations against the results from interferometry measurements. With a maximum uncertainty by a factor of two in the free electron density, the simulations indicate extremum values for the temperature by ± 40 eV, which are thus taken as its maximum uncertainty.

- p.8, paragraph 4: *The last sentence ends "...with a sufficient precision to disprove these theories." I recommend qualifying this statement with something like "in this parameter range", since the*

these other theories still seem adequate in the high-velocity region for which they were originally designed, and indeed are stated to be appropriate in this regime on page 3 of this manuscript.

"in the studied parameter range" was added (page 10, though not at the same line as before, as the text has been reformulated to a large extent) .

- p.8, last paragraph: the first sentence states these measurements are used to discriminate theories in "dense plasmas", and two sentences later the plasma conditions are instead described as "ideal and nondegenerate". These statements are confusing if not inconsistent. Based on the parameters reported in equation 2, the conditions seem to be weakly-coupled and non-degenerate, which are characteristics of low-density plasmas.

The qualification "dense plasmas" was indeed misleading and subjective. Instead, we wrote "highly ionized plasma" and we kept the qualification "nearly ideal and nondegenerate", which is accurate (page 10).

- p.9 "setup" paragraph 1: I recommend also stating the laser irradiance on target.

The laser irradiance was added in the "Setup" section in the methods (page 11), and it is of about 5.10^{11} W/cm².

- p.10 "targets" paragraph 1: Does the initial areal density error range represent the range among different targets, or the precision of an individual target?

The uncertainty in the target areal density was indeed not clearly indicated, moreover we now corrected the error on this areal density to be of 1%. We also indicated that the different thicknesses are included in a $\pm 5\mu\text{g}/\text{cm}^2$ interval, and each of them is known with a precision of 1%. The error analysis on the energy loss measurements has been accordingly modified, and other contributions have also been quantified in a more appropriate way.

- There is significant temporal evolution of the plasma over the rather long (> 5 ns) arrival interval of the ion bunch. How this folds in to the analysis should be described explicitly (if briefly) in the body text. As it stands, there is only a passing remark in the caption to fig 2 regarding this step of the analysis.

We added the sentence "Each energy-loss value is averaged over the plasma parameters in a 5.5 ns range corresponding to the experimental bunch length." on page 8 of the main text to mention the influence of the bunch duration in the simulation.

G. References

References are adequate.

H. Clarity and context

Although much of the narrative needs significant revision to improve the clarity of the data analysis steps, the context given in the abstract, introduction, and conclusions are reasonably clear. A remark on page 9, "This finding has also strong implications for other transport and relaxation properties where similar electron-collisions play an important role," may be worth further elaboration, as these as-yet unidentified implications could help place stopping power in better

context within the scope of Coulomb-mediated transport and relaxation properties in general.

We extended the sentence "This finding has also strong implications for other transport and relaxation properties where similar electron collisions play an important role" at the end of the main text (page 11) to "This finding has also strong implications for other transport and relaxation properties like temperature equilibration or thermal and electrical conductivity where close collisions play a similarly important role." and we added a citation to a publication by Gericke et al. treating temperature equilibration in dense plasmas as well as a citation to a work by Haun et al. about conductivity in dense plasmas. In this way, we suggest the potential consequences of our results on other plasma properties.

Finally, I recommend the native english speaking co-authors review the text for proper articles and awkward phrasing. Examples include one in the first paragraph, "...allows to clearly rule out..." and in the last paragraph before the methods section "...in addition of their direct relevance..."

Conclusion

We are grateful to Reviewer 3 for his detailed report, his praise on the work's quality and his recommendation for publication after the relevant changes have been performed. We also thank him for his numerous and useful comments to improve the clarity and the scientific quality of the work. All the points mentioned by the Referee have been addressed and we believe that this has led to a significant improvement in the quality of the manuscript.

Reviewers' comments:

Reviewer #1 (Remarks to the Author):

In my opinion, all points raised by myself as well as other reviewers are clearly addressed. In the revised manuscript, the theoretical discussions are enhanced, the experimental data are added, and the corresponding analysis is also reinforced. Briefly, the revised manuscript is improved sufficiently. This work provides a fundamental understanding of the stopping power of ion in the plasmas. So, I recommend publication of the revised manuscript in Nature Communications.

Reviewer #2 (Remarks to the Author):

Review of revised manuscript NCOMMS-16-16897-T ("Experimental discrimination of ion stopping models near the Bragg peak in highly ionized matter") by W Cayzac et al. Overall I am pleased with the revisions made to the manuscript. The authors seem to have done good job addressing most of the referees concerns. I only have one more issue that I think needs to be addressed. As dE/dx around the Bragg peak scales with $1/T_e$, it seems to me that the Li-Petrasso model would do as well as the other models, if not a better job, describing the data if T_e was in fact 190 eV instead 150 eV. At 190 eV, which is within the quoted error of ± 40 eV, the modeled dE/dx and energy loss would be $\sim 25\%$ lower. With this in mind and the fact that T_e value used in the analysis is solely based on modeling, I am concerned that the author's statement about 'experimental discrimination of ion stopping models...' is too strong. That said, I'd like to either see a stronger discussion that better refutes the likelihood of 190 eV or weaken the conclusions in the title and text throughout. After they've had addressed this issue, I am perfectly happy to see this work published in Nature Communication.

Responses to the Reviewers' comments

The Reviewers' comments are written in blue italic, while our responses appear below.

Reviewer #1 (Remarks to the Author):

In my opinion, all points raised by myself as well as other reviewers are clearly addressed. In the revised manuscript, the theoretical discussions are enhanced, the experimental data are added, and the corresponding analysis is also reinforced. Briefly, the revised manuscript is improved sufficiently. This work provides a fundamental understanding of the stopping power of ion in the plasmas. So, I recommend publication of the revised manuscript in Nature Communications.

We are sincerely grateful for the Reviewer's praise of our work and for his recommendation of publication in Nature Communications.

Reviewer #2 (Remarks to the Author):

Review of revised manuscript NCOMMS-16-16897-T (“Experimental discrimination of ion stopping models near the Bragg peak in highly ionized matter”) by W Cayzac et al. Overall I am pleased with the revisions made to the manuscript. The authors seem to have done good job addressing most of the referees concerns. I only have one more issue that I think needs to be addressed. As dE/dx around the Bragg peak scales with $1/Te$, it seems to me that the Li-Petrasso model would do as well as the other models, if not a better job, describing the data if Te was in fact 190 eV instead 150 eV. At 190 eV, which is within the quoted error of ± 40 eV, the modeled dE/dx and energy loss would be $\sim 25\%$ lower. With this in mind and the fact that Te value used in the analysis is solely based on modeling, I am concerned that the author’s statement about ‘experimental discrimination of ion stopping models...’ is too strong. That said, I’d like to either see a stronger discussion that better refutes the likelihood of 190 eV or weaken the conclusions in the title and text throughout. After they’ve had addressed this issue, I am perfectly happy to see this work published in Nature Communication.

We thank Reviewer #2 for his praise of the revised version of the manuscript and for his additional remark that enables us to improve the clarity of the discussion and of our conclusions.

The final issue raised by Referee 2 is already answered in our panel d) of Fig.3. Here, we do not rely on a scaling of the stopping power as suggested by the referee but we perform the full calculation of the Li-Petrasso and of the T-matrix models including the largest possible range of input densities and temperatures. When compared to our measurements, the band of predictions of the Li-Petrasso model is clearly ruled out, while the band of predictions of the T-matrix is consistent with our experimental data. The calculation entitled "LP min" corresponds, at the same time, to the maximum possible plasma temperature (+ 40 eV) and maximum possible plasma density (twice the nominal density). Hence, this case leads to the smallest possible energy loss (about 15% smaller than in the nominal case). However, even this extreme case is still clearly outside the 1σ experimental error bars.

Hence, the results presented in figure 3d support our claim having experimentally distinguished between the Li-Petrasso and the T-Matrix model in spite of the significant uncertainty in the plasma temperature.

REVIEWERS' COMMENTS:

Reviewer #2 (Remarks to the Author):

Yes, agreed. A higher T_e would in fact improve the chi-square between the data and TM modeling. Although this is also the case for the LP modeling, the LP modeling can be ruled out with certain level of confidence. I stand corrected, and I recommend publication of the revised manuscript in Nature Communications.